# Antioxidant and Antiapoptotic Effects of a *Turraea fischeri* Leaf Extract on Cryopreserved Goat Sperm

**DOI:** 10.3390/ani11102840

**Published:** 2021-09-29

**Authors:** Soha A. Hassan, Wael A. Khalil, Mahmoud A. E. Hassan, Ahmed I. Yousif, Omar M. Sabry, Michael Wink, Mansour Sobeh

**Affiliations:** 1Basic Science Department, Faculty of Dentistry, October 6 University, Cairo 12566, Egypt; sohaahmed1999@yahoo.com; 2Department of Animal Production, Faculty of Agriculture, Mansoura University, Mansoura 35516, Egypt; 3Agriculture Research Center, Animal Production Research Institute, Ministry of Agriculture, Dokki, Giza 12619, Egypt; m.hassan55213@gmail.com (M.A.E.H.); ahmedyousif2788@yahoo.com (A.I.Y.); 4Department of Pharmacognosy, College of Pharmacy, Cairo University, Cairo 11562, Egypt; omar.sabry@pharma.cu.edu.eg; 5Institute of Pharmacy and Molecular Biotechnology, Heidelberg University, 69120 Heidelberg, Germany; wink@uni-heidelberg.de; 6AgroBioSciences Research, Mohammed VI Polytechnic University, Lot 660–Hay MoulayRachid, Ben Guerir 43150, Morocco

**Keywords:** *Turraea fischeri*, polyphenolics, semen cryopreservation, sperm ultrastructure, antioxidant biomarker, apoptosis

## Abstract

**Simple Summary:**

The excessive production of reactive oxygen species (ROS) in cryopreservation and post-thawing affects sperm quality and subsequent fertilizing ability. Antioxidants of natural origin, such as plant extracts, rich in flavonoid and phenolic compounds, are of special interest in scavenging ROS. The supplementation of goat semen extender with 375 µg/mL *T. fischeri* leaf extract improved the functional and ultrastructural characteristics of cryopreserved sperm by maintaining antioxidant capacity, thus preventing membrane injury and reducing apoptosis.

**Abstract:**

This study evaluated the efficacy of *Turraea fischeri* leaf extract for maintaining the viability of cryopreserved goat sperm. Ejaculated semen was collected from 5 mature Baladi bucks (50–60 kg, 2–4 years of age) and those samples with mass motility ≥ 70% and sperm concentration ≥ 2.5 × 10^9^/mL were selected, pooled, and divided into 4 aliquots. Each aliquot was diluted in Tris-citric-soybean lecithin extender containing a different concentration of *T. fischeri* leaf extract (0, 125, 250, or 375 µg/mL). Treated semen samples were cooled to 5 °C, transferred to 0.25-mL French straws, and stored in liquid nitrogen (LN_2_) at −196 °C. After thawing, membrane integrity was examined by transmission electron microscopy, apoptotic activity by Annexin/propidium iodide staining and flow cytometry, and both enzyme activities and antioxidant capacity by spectroscopic assays. The leaf extract at 375 µg/mL significantly improved semen quality as indicated by enhanced total antioxidant capacity, reduced H_2_O_2_ concentration, a greater proportion of structurally intact motile sperm, and concomitant reductions in apoptosis and necrosis. The extract also significantly increased the proportion of sperm with a contiguous plasma membrane and intact acrosome (*p* < 0.05). Furthermore, LC-MS revealed numerous secondary metabolites in the extract that may contribute to sperm cryopreservation.

## 1. Introduction

Artificial insemination (AI) is used widely in agriculture to optimize and spread commercially valuable genetic traits, including in goats. Effective cryopreservation of semen samples is critical for efficient AI. However, cryopreservation can reduce goat sperm quality, motility, and viability, resulting in lower fertility rates [1]. The reduced semen quality results in part from cellular oxidative stress, which causes the peroxidation of unsaturated fatty acids in the biomembrane. In healthy mammalian sperm cells, total antioxidant capacity (TAC) and reactive oxygen species (ROS) production remain in balance [2], but an increase in ROS production, a decrease in TAC, or both during cryopreservation can result in reduced motility and cell death [3]. Excessive ROS production during cryopreservation and after thawing not only degrade cellular membranes but may also damage DNA [4], further enhancing dysfunction [5]. To reduce oxidative stress, exogenous antioxidants are frequently added to cryopreservative solutions to sustain semen quality prior to AI [6]. Several studies have confirmed ROS scavenging capacity of various medicinal plant extracts to improve sperm motility and increase fertility rates [7,8,9,10].

The genus *Turraea* L., a genus of the Meliaceae (mahogany) family of tropical flowering trees and shrubs, includes 70 species distributed throughout Africa and Asia [11,12]. *Turraea fischeri* is widely used in east Africa to treat stomachache and infertility [13]. Previous studies have also documented antioxidant and hepatoprotective properties [14]. Furthermore, biochemical analyses of *T. fischeri* extracts have identified numerous potentially bioactive secondary metabolites such as limonoids as well as various sterols and flavonoids with antioxidant and anti-inflammatory activities [15].

To date, there are no studies available on the effect of the leaf extract from *T. fischeri* in semen extenders on semen cryopreservation. In the current study, the chemical profile of a leaf methanol extract from *T. fischeri* was characterized using HPLC-MS/MS, and its antioxidant activity was measured by in vitro spectroscopic assays. The effects of *T. fischeri* leaf extract on goat sperm viability, motility, and morphology, and antioxidant capacity were then examined after thawing by vital staining under light microscopy, flow cyto-metry following Annexin/propidium iodide staining, various spectroscopic assays, and transmission electron microscopy.

## 2. Materials and Methods

### 2.1. Plant Material, Extraction, LC-MS and Antioxidant Acctivities

Leaves of *Turraea fischeri* were collected from the Lupaga Site in Shinyanga, Tanzania, and stored at the Institute of Pharmacy and Molecular Biotechnology, Heidelberg University, under accession number P7336 [16].

The dried and ground leaves (100 g) were extracted three times in 100% methanol (3 × 500 mL) at ambient temperature and the three extracts combined. Pooled extracts were then filtered and dried under a vacuum at 40 °C. The obtained residue was lyophilized, yielding a fine dried powder (15 g). LC-MS analyses and in vitro antioxidant activities were performed according to Sobeh et al. [14]. Detailed methods are included in the Appendix A.

### 2.2. Animals

Semen samples were collected from 5 mature, fertile Baladi bucks (50–60 kg LBW; 2–4 years of age) at the Animal Production Research Station, El-Karada, Kafrelsheikh, Animal Production Research Institute (APRI), Agricultural Research Center, Ministry of Agriculture, Egypt, using an artificial vagina. Sample collection was conducted in cooperation with the Physiology and Biotechnology Laboratory, Animal Production Department, Faculty of Agriculture, Mansoura University, Egypt, according to animal welfare guidelines of Mansoura University.

### 2.3. Animal Management

All 5 bucks were raised under the same environmental conditions. Feeding requirements were calculated according to the recommendations of Animal Production Research Institute, Ministry of Agriculture, Egypt. Each buck was fed 1.0 kg/day concentrate feed mixture containing 14% crude protein and 70% total digestible nutrients, plus 1.25 kg/day berseem hay from August to November or 5 kg/day Egyptian fresh berseem clover (*Trifolium alexandrinum*) from December to February. Animals had free access to trace mineralized salt and drinking water at all times.

### 2.4. Collection of Semen

Semen was collected from each buck by artificial vagina once weekly before feeding at 7–8 a.m. for five consecutive weeks. In total, 25 samples were obtained. Samples were transferred immediately into a water bath at 37 °C and only those with mass motility ≥ 70% and sperm concentration of at least 2.5 × 10^9^/mL were retained for experiments. Samples were then pooled and divided into 4 aliquots for each treatment group.

### 2.5. Preparation of Extender

The Tris-citric-soybean lecithin extender contained 3.025 g/dL Tris (Sigma Chemical Co., St. Louis, MO, USA), 1.66 g/dL citric acid monohydrate (Sigma, Darmstadt, Germany), 1.25 g/dL glucose (Sigma Aldrich, St. Louis, MO, USA), 5% glycerol (Honeywell, Regen, Germany), 1% soybean lecithin (L-a-phosphatidylcholine, LAB: product number MC041), 100 IU/mL penicillin, and 100 µg/mL streptomycin. The components were mixed in a water bath at 37 °C and adjusted to 300 mOsm/kg in H_2_O (pH 6.8) before addition of extract.

### 2.6. Cryopreservation

Pooled semen was diluted 1:10 in extender (*v*/*v*) containing the indicated extract concentration (0, 125, 250, and 375 µg/mL) and adjusted to a final sperm concentration of 2.5 × 10^8^/mL. The mixture was then gradually cooled from 37 °C to 5 °C over 4 h (equilibration period) and transferred to 0.25-mL French straws (IMV Technologies, L’Aigle, France) for cryopreservation. Straws were first exposed to liquid nitrogen vapor for 10 min and then immersed in liquid nitrogen at −196 °C.

### 2.7. Thawing

After one month, the straws were thawed at 37 °C for 30 s in a water bath, and the various assessments conducted immediately.

### 2.8. Semen Evaluation

#### 2.8.1. Progressive Motility

The proportion of sperm cells showing progressive motility was examined under a phase-contrast microscope (DM 500, Leica, Switzerland) with heated stage set to 37 °C. Briefly, aliquots of diluted sperm (10 µL) were placed on pre-warmed glass slides and sealed with coverslips. A total of 200 spermatozoa/slide from 3 randomly chosen fields were counted by the same investigator and the mean proportion (%) recorded.

#### 2.8.2. Viability

Semen samples were double-stained with a mixture of 5% eosin (vital stain) and 10% nigrosin (background stain) to estimate the live: dead ratio [17]. The live: dead ratio was calculated by counting the unstained head area among 300 sperm at high magnification (400×) using a light microscope.

#### 2.8.3. Gross Structural Abnormalities

Abnormalities in gross structure were assessed in 300 sperm cells during viability measurements using a light microscope. The following criteria were considered: (i) tail defects (abnormal tails), (ii) abnormal heads, and (iii) cytoplasmic droplets [18].

#### 2.8.4. Plasma Membrane Integrity

The hypo-osmotic swelling test (HOS-t) was used to assess plasma membrane (PM) integrity according to a previously described protocol [19]. Briefly, semen (50 µL) was incubated at 37 °C for 30 min in a hypo-osmotic solution (500 µL at 75 mOsm/kg) containing fructose (6.75 g/L) and sodium citrate (3.67 g/L) in H_2_O. A sample of the mixture was placed on a slide and covered with a coverslip. The number of spermatozoa with coiled or swollen tails (indicative of intact membranes under hypoosmotic conditions) among 300 sperm per slide was counted in each sample at 400× under phase-contrast microscopy.

#### 2.8.5. Antioxidant Capacity and Enzyme Activities

The following biochemical parameters were measured in post-thawed extender: total antioxidant capacity (TAC, linearity up to 2 mM/L) [20], hydrogen peroxide concentration (H_2_O_2_, linearity up to 1.5 mM/L) [21], lactic dehydrogenase (LDH, linearity up to 1700 units/L) activity [22], aspartate transaminase (AST, linearity up to 150 units/mL) activity, and alanine transaminase (ALT, linearity up to 120 units/mL) activity [23]. All measurements were performed using a spectrophotometer (Spectro UV-VIS Auto, UV-2602, Labomed, Los Angeles, CA, USA) and commercial kits (Biodiagnostic, Giza, Egypt) according to the manufacturer’s instructions.

#### 2.8.6. Apoptosis and Necrosis

Semen samples were stained with Annexin-V (AV, calcium-dependent probe) for tracking phosphatidylserine (PS) externalization in the membrane and propidium iodide (PI) as an indicator of genomic DNA exposure using a commercial PS Detection Kit (IQP, Groningen, The Netherlands) according to the manufacturer’s instructions. Briefly, semen samples were thawed and washed twice by centrifugation (300× *g* for 10 min at 4 °C) with phosphate-buffered saline. After the second centrifugation, the supernatant was removed, and the sperm pellet resuspended in binding buffer at 1 × 10^6^ sperm cells/mL. Then, 100 μL of semen sample was transferred to culture tubes (5 mL) containing 5 μL AV (fluorescein isothiocyanate, FITC label, BD Biosciences, San Jose, CA, USA) and 5 μL PI (BD Biosciences). The mixed suspension was then incubated in the dark at room temperature (25 °C) for 15 min, followed by the addition of 400 µL binding buffer to each tube. Staining patterns were then evaluated by flow cytometry using an Accuri C6 Cytometer and Accuri C6 software (BD Biosciences) [24]. Cells negative for both AV and PI staining (A−/PI−) were classified as viable, those positive for AV and negative for PI (A+/PI−) as early apoptotic, those positive for both AV and PI (A+/PI+) as apoptotic, and those negative for AV and positive for PI (A−/PI+) as necrotic.

#### 2.8.7. Ultrastructure

Sperm ultrastructure was examined by transmission electron microscopy (TEM) as described [25] with some modifications. Briefly, 500 μL of each semen sample was centrifuged and resuspended in cold (4 °C) fixative solution (2.5% glutaraldehyde in phosphate buffer) for 2 h. Samples were then washed and post-fixed in osmium tetroxide (1%) for 90 min at room temperature, dehydrated, cleared in gradient ethanol and propylene oxide, and embedded in Epon 812 (Fluka Chemie, AG, Buchs, Switzerland). Ultrathin sections (60–70 nm) were prepared using glass knives and observed using a JEOL-JEM 2100 TEM at 80 kV. Changes in PM and acrosome ultrastructure were examined from 300 sperm per sample.

### 2.9. Statistical Analysis

Arcsine transformation was performed before statistical analyses because that helps in dealing with percentage values for semen characteristics including progressive motility, viability, membrane integrity, acrosome integrity, structural abnormality, sperm viability by Annexin-V, plasma membrane integrity, acrosomal ultrastructure. Treatment group means were compared by one-way analysis of variance (ANOVA) [26] and Duncan’s multiple range tests [27]. A *p* < 0.05 was considered statistically significant for all tests.

## 3. Results

### 3.1. Chemical Composition and In Vitro Antioxidant Activities

Analysis of a leaf methanol extract from *T. fischeri* by HPLC-MS/MS tentatively identified 17 compounds, including particularly high concentrations of phenolic acids (e.g., compounds 2, 3, and 4), flavonoids (compounds 10–17), and corresponding glycoside derivatives (Table 1). On the other hand, the bark extract from the same plant was rich in 20 secondary metabolites belonging to cinchonains and phenylpropanoid-substituted catechin [14]. The extract exhibited substantial antioxidant activity in two commonly used assays; FRAP and DPPH (Table 2). The observed results might be attributed to the existence of several phenolic acids (*p*-coumaroylquinic acid, feruloylquinic acid and caffeoylquinic acid) and flavonoids (Quercetin rutinoside, quercetin glucoside, kaempferol rutinoside, kaempferol glucoside and isorhamnetin glucoside). Comparable activities were reported from the bark extract [14].

### 3.2. Effects of T. fischeri Leaf Extract on the Viability, Morphology, Function, and Membrane Integrity of Thawed Cryopreserved Goat Sperm

#### 3.2.1. Effects on Sperm Motility, Viability, and Membrane Integrity

Post-thaw motility, viability, and membrane integrity were all significantly (*p* < 0.05) improved by addition of *T. fischeri* leaf extract (TFLE) to semen extender during cryopreservation, and the degree of improvement increased with TFLE concentration compared to semen extender alone, Table 3. Alternatively, acrosome integrity and the proportion of abnormal sperm were not affected significantly by TFLE (*p* > 0.05, Table 3).

#### 3.2.2. Effects on Apoptosis and Necrosis Rates

The proportion of viable sperm was significantly higher in TFLE-treated cryopreserved semen than control cryopreserved semen (*p* < 0.05), and the viable fraction increased progressively with TFLE concentration (Table 4). Consistent with this dose-dependent improvement in viability, the proportions of early apoptotic, total apoptotic, and necrotic spermatozoa decreased progressively with TFLE dose compared to controls. Further, at 375 µg/mL TFLE, the proportion of necrotic sperm was significantly lower than at all other concentrations.

#### 3.2.3. Effects on Membrane Ultrastructure

Examination of sperm cell ultrastructure by TEM revealed that the plasma membrane (PM) was damaged by cryopreservation in semen extender alone, while supplementation with TFLE dose-dependently improved sperm ultrastructure after thawing. Table 5 summarizes the proportions of sperm cells in each treatment group demonstrating an intact, slightly swollen, swollen, or disrupted PM. Consistent with viability analyses, there was a TFLE dose-dependent increase in the portion of cells with an intact PM and decreases in the proportions with slightly swollen and swollen PM.

Figure 1A–G illustrates the different ultrastructural abnormalities resulting from cryopreservation. Four injury patterns were defined according to the degree of PM damage [28]. (i) Sperm with intact PM exhibited a normal head region with an intact acrosome (IA) and contiguous PM tightly surrounding the acrosomal ground substance (Figure 1A–C). In addition, the mid-sectional region of healthy sperm contained a contiguous mitochondrial sheath (MS) completely enclosing morphologically typical mitochondria. The axoneme also exhibited the normal 9 + 2 arrangement of microtubules (Figure 1B,C). (ii) Sperm with slightly damaged PMs exhibited an acrosomal physiological reaction (AR) characterized by the initial formation of small vesicles under a dilated and slightly separated plasma membrane (SP) (Figure 1A,B). (iii) In the third category, sperm exhibited a swollen PM (S) with a wavy appearance, and mitochondrial sheathes were also dilated (Figure 1D,F). (iv) Finally, another fraction of sperm showed discontinuous or disintegrated (DS) PMs, damaged membranes (DMs) around mitochondria, and various mitochondrial ultrastructural abnormalities. The axoneme structure at this stage also showed an abnormal microtubule arrangement. The distribution of these categories differed among treatment groups, particularly at higher TFLE concentrations. In the 375 µg/mL group, there was a significantly greater proportion of cells with an IA and a smaller proportion with atypical acrosomes (*p* < 0.05). There was also a trend for lower frequencies of typical AR and lost acrosome in TFLE-treated groups compared to controls cryopreserved in extender only (Table 6).

#### 3.2.4. Effects on Oxidative Biomarkers and Enzyme Activity

Addition of TFLE also dose-dependently enhanced insignificantly post-thaw TAC and reduced significantly H_2_O_2_ (*p* < 0.05, Table 7). Alternatively, TFLE had no significant effects on LDH, AST, and ALT activities as shown in Table 7.

## 4. Discussion

Supplementation of goat semen extender with 375 µg/mL TFLE improved the functional and ultrastructural characteristics of cryopreserved sperm by maintaining antioxidant capacity, thereby preventing membrane injury and reducing apoptosis.

Semen cryopreserved in semen extender alone (control group) demonstrated the lowest proportion of viable sperm compared to samples preserved with extender containing varying concentrations of the leaf extract. Freezing and thawing can damage cellular membranes, reducing sperm number and quality as shown by flow cytometry and TEM. Our results are in accord with previous studies demonstrating the harmful effects of thawing on sperm structure and function. Oxidative stress during cryopreservation reduces the reproductive potential of semen by impairing sperm motility, reducing mitochondrial activity, damaging DNA, and activating apoptotic pathways [8,9,29]. Thus, supplementation of semen extender with antioxidants prior to cryopreservation is recommended to facilitate efficient goat breeding [30]. Here, we demonstrate improved cryopreservation using a natural plant extract containing multiple bioactive agents with known beneficial effects against cellular stress.

Freezing and thawing induced apoptosis as evidenced by Annexin staining, which reveals the translocation of phosphatidylserine from the inner to the outer PM layer. Further, some sperm cells became necrotic during cryopreservation as evidenced by PI staining [31]. Both apoptosis and necrosis are associated with loss of PM integrity, which is necessary to maintain sperm function within the female reproductive tract [32]. During cryopreservation, rearrangement of membrane lipids alters fluidity and increases susceptibility to disruption, which then induces further cellular damage and ultimately death [33]. These pathological effects are manifested by changes in sperm morphology during the freezing-thawing process [34]. High concentrations of polyunsaturated fatty acids (PUSFAs) such as arachidonic and docosahexaenoic acids in the PM increase the vulnerability to ROS-induced peroxidative damage and membrane dysfunction [35]. Further, oxidative injury may be spread throughout the spermatozoa population by a subset of cells overproducing ROS [36], leading to generally reduced mitochondrial metabolic activity, motility, and viability [37]. Maintenance of cell membrane integrity and mitochondrial function under oxidative stress are thus essential for successful fertilization using cryopreserved semen [33].

Following cryopreservation, damage to the PM and acrosomal cap was predominantly observed in the head region, in accordance with previous observations of human sperm [28]. Membrane swelling is most probably caused by changes in the extracellular osmotic pressure during freezing and thawing, causing cells to accumulate or lose water. The sperm PM is known to mediate the exchange of sodium, potassium [38,39], and calcium [40], and these ion fluxes regulate motility and mitochondrial function as well as osmotic balance. An intact PM is also necessary for fusion with the outer acrosomal membrane and induction of the acrosome reaction [41]. Acrosomal integrity is also essential for fertilization as this organelle contains hydrolytic enzymes such as hyaluronidase, acrosin, and esterases required for lysis of the zona pellucida and penetration of the oocyte corona radiata [42]. Freezing and thawing significantly increased the number of sperm cells with atypical acrosomal structure, which has previously been attributed to degeneration and apoptosis [43].

The addition of TFLE to sperm extender dose-dependently increased the TAC of goat semen and reduced the concentration of H_2_O_2_, a major ROS generator. On the other hand, TFLE had little effect on the activities of LDH, AST, and ALT. The dose-dependent increase in TAC was strongly associated with the progressive rise in sperm cell viability and the decreases in apoptosis, necrosis, and structural abnormalities. Consistent with these findings, Salimi, et al. [44] reported positive correlations between TAC and both sperm motility and normal sperm morphology, while Pahune, et al. [45] observed positive correlations between TAC and multiple seminogram parameters including sperm concentration, sperm motility, and normal sperm morphology. Collectively, these findings suggest that an imbalance between TAC and ROS production is a major contributor to impaired sperm function following cryopreservation [46].

Mitochondrial enzymatic activities in human spermatozoa are strongly correlated with motility [47]. Aspartate transaminase (AST) and ALT are essential for metabolic processes that provide energy for sperm survival, motility, and fertility [48], and so are good indicators of sperm membrane stability and semen quality [49]. An increase in spermatozoa damage within the liquid storage medium results in an elevated concentration of transaminase enzymes [50]. Indeed, AST and ALT activities were slightly higher in control samples than samples containing 375 µg/mL TFLE, although the difference did not reach statistical significance.

TFLE contains secondary metabolites such as malic acid, quercetin, and kaempferol that may contribute to these improved functional and structural characteristics. Indeed, malic acid decreases the accumulation of ROS and enhances the glutathione cycle by regulating various endogenous antioxidant pathways [51,52], while the flavonoids can directly scavenge ROS, thereby resisting oxidative damage during cryopreservation [53]. Finally, kaempferol is a flavonoid compound with potent activity against inflammation caused by oxidative stress [54].

The effect of TFLE on semen cryopreservation was stronger compared with other plant extracts such as *Albizia harveyi* leaf extract in bull [8], *Entada abyssinica* bark extract in ram [9], and nanoformulations of mint, thyme, and curcumin in goat [10]. All these extracts enhanced semen preservability and sperm characteristics after freezing and thawing.

TFLE is rich in phenolic and flavonoid compounds, which have antioxidant properties. This is evident in the antioxidant activity of the extract via DPPH, FRAP, and TPC assays or by reducing the concentration of hydrogen peroxide in the semen extender after thawing. This can be explained by the ability of polyphenolic compounds to scavenge reactive oxygen species such as superoxide anion radicals and hydroxyl radicals, thus interrupting free radical chain reaction [55]. Therefore, we expect the same effect in preserving the semen in other species.

## 5. Conclusions

The addition of 375 µg/mL TFLE to Tris-soybean lecithin extender significantly improved the cryopreservation of goat semen as evidenced by a greater proportion of cells retaining robust motility, viability (low apoptosis rate), and normal ultrastructure after thawing. These benefits were associated with elevation of semen antioxidant capacity. The efficacy of the extract in artificial insemination needs to be studied in more detail.

## Figures and Tables

**Figure 1 animals-11-02840-f001:**
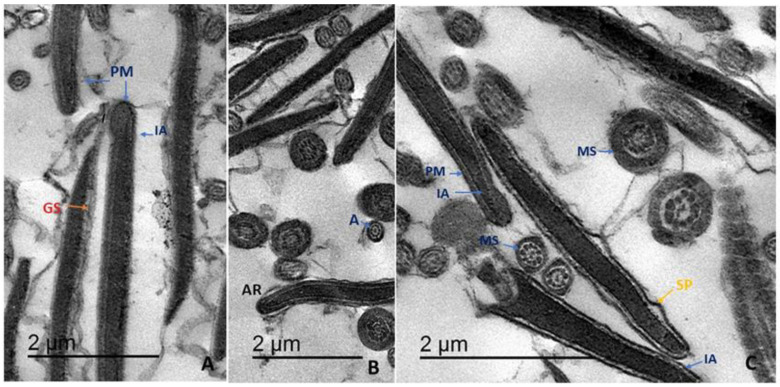
Addition of *T.*
*fischeri* leaf extract to semen extender during cryopreservation significantly improved post-thaw sperm ultrastructure. (**A**–**C**) Intact sperm with structurally intact acrosomes (IA) completely enclosed by contiguous plasma membranes (PM). The PM appears continuous alongside the nucleus in longitudinal sections and around the mitochondrial sheath (MS) in cross-sections. Few sperm cells exhibited a separated plasma membrane (SP) or diffusion of ground substance (GS) under a detached PM. (**D**) Damaged sperm showing a swollen ‘wavy’ PM leaving a large space around the nucleus (S). (**E**) Cross-sections of the tail region showing swollen PM containing cytoplasmic residue (CR) and mitochondrial sheathes enveloping damaged mitochondria (DM). (**F**) Damaged acrosomal cap (DAC), dilated PM in the tail region (Arrows), and cytoplasmic residue (CR). (**G**) Discontinuous (DS) PM in longitudinal and cross-sections of different sperm regions.

**Table 1 animals-11-02840-t001:** Characterization of secondary metabolites of *T. fischeri* by HPLC-MS/MS analyses. Precursor ions and corresponding fragment ions.

No.	Rt	M-H	MS/MS	Proposed Compound
1	1.49	133	115	Malic acid
2	2.34	353	191, 179	Caffeoylquinic acid
3	3.11	337	173	*p*-Coumaroylquinic acid
4	4.39	367	193, 173	Feruloylquinic acid
5	4.84	385	265, 137	Hydroxybenzoic acid rhamnoside derivative
6	9.03	411	265, 163, 119	*p*-Coumaric acid rhamnoside derivative
7	10.69	441	265, 193	Ferulic acid glucuronide derivative
8	12.44	505	463, 301, 179	Methyl quercetin glucoside
9	12.96	549	505, 301, 179	Dimethyl ether quercetin glucoside
10	21.72	741	609, 301, 271	Quercetin pentosyl-rutinoside
11	23.2	609	301, 179	Quercetin rutinoside
12	24.25	609	301, 179	Quercetin rutinoside
13	25.94	463	301, 179, 151	Quercetin glucoside
14	28.69	593	285	Kaempferol rutinoside
15	29.77	447	285	Kaempferol glucoside
16	31.2	477	315, 299	Isorhamnetin glucoside
17	32.34	477	315, 299	Isorhamnetin glucoside

Rt: retention time, M-H: pseudo molecular ion in the negative ion mode, MS/MS: mass fragmentation pattern.

**Table 2 animals-11-02840-t002:** Antioxidant activities of a leaf methanol extract from *T. fischeri* as measured by FRAP, DPPH, and TPC assays. GAE = gallic acid equivalents.

Sample	FRAP	DPPH	TPC
	(mM FeSO_4_ Equivalent/mg Extract)	(EC_50_ µg/mL)	mg GAE/g Extract
Leaf extract	9.84 ± 0.19	50.3 ± 3.5	236
Ascorbic acid	-	2.92 ± 0.29	-
Quercetin	24.04 ± 1.23	-	-

Ascorbic acid and quercetin are positive controls. DPPH, 2,2-diphenyl-1-picryl-hydrazyl-hydrate; FRAP, ferric-reducing antioxidant power assay; TPC, total phenolic content.

**Table 3 animals-11-02840-t003:** Effects of *T. fischeri* leaf extract on cryopreserved goat sperm characteristics (means ± SE, *n* = 5).

Concentration	Sperm Characteristic (% of Cells Examined)				
	Progressive Motility	Viability	Membrane Integrity	Acrosome Integrity	Structural Abnormality
Control	47.0 ± 2.55 ^b^	45.0 ± 2.05 ^b^	39.4 ± 0.87 ^b^	86.8 ± 1.71	13.4 ± 1.33
Extract 125 µg/mL	54.0 ± 1.87 ^a^	48.6 ± 1.94 ^a,b^	48.6 ± 1.47 ^a^	88.8 ± 1.50	14.4 ± 1.03
Extract 250 µg/mL	59.0 ± 1.87 ^a^	53.8 ± 1.16 ^a^	51.6 ± 2.54 ^a^	86.4 ± 1.63	12.8 ± 0.37
Extract 375 µg/mL	60.0 ± 1.58 ^a^	52.0 ± 1.58 ^a^	48.8 ± 1.74 ^a^	87.6 ± 1.21	14.0 ± 1.70

^a^ and ^b^: Values in the same column with different superscripts are significantly different at *p* < 0.05.

**Table 4 animals-11-02840-t004:** Effects of *T. fischeri* leaf extract on goat sperm viability (means ± SE, *n* = 3).

Concentration	Viable (%)	Early Apoptotic (%)	Apoptotic (%)	Necrotic (%)
	(A−/PI−)	(A+/PI−)	(A+/PI+)	(A−/PI+)
Control	39.7 ± 0.49 ^d^	25.1 ± 0.69 ^a^	31.8 ± 0.03 ^a^	3.4 ± 0.23 ^c^
Extract 125 µg/mL	51.9 ± 0.06 ^c^	13.0 ± 0.15 ^c^	25.2 ± 0.17 ^b^	9.9 ± 0.03 ^a^
Extract 250 µg/mL	58.9 ± 0.55 ^b^	12.0 ± 0.30 ^c^	20.6 ± 0.40 ^c^	8.5 ± 0.17 ^b^
Extract 375 µg/mL	62.4 ± 0.55 ^a^	19.4 ± 0.55 ^b^	17.3 ± 0.03 ^d^	0.9 ± 0.03 ^d^

^a^, ^b^, ^c^ and ^d^: Values in the same column with different superscripts are significantly different at *p* < 0.05.

**Table 5 animals-11-02840-t005:** Effect of *T. fischeri* leaf extract on sperm plasma membrane (PM) integrity (means ± SE, *n* = 3).

Concentration (mg/mL)	Intact PM (%)	Slightly Swollen PM (%)	Swollen PM (%)	Lost PM (%)
Control	39.3 ± 0.88 ^c^	17.3 ± 0.88 ^a^	33.7 ± 0.88 ^a^	9.7 ± 0.88
Extract 125 µg/mL	43.0 ± 1.15 ^c^	19.0 ± 1.00 ^a^	30.0 ± 1.53 ^b^	8.0 ± 1.15
Extract 250 µg/mL	54.3 ± 2.33 ^b^	11.0 ± 1.53 ^b^	24.7 ± 0.88 ^c^	10.0 ± 1.53
Extract 375 µg/mL	64.4 ± 2.33 ^a^	6.0 ± 2.31 ^b^	21.3 ± 0.88 ^c^	8.3 ± 1.20

^a^, ^b^ and ^c^: Values in the same column with different superscripts are significantly different at *p* < 0.05.

**Table 6 animals-11-02840-t006:** Effect of *T. fischeri* leaf extract on acrosomal ultrastructure (means ± SE, *n* = 3).

Concentration (mg/mL)	Intact Acrosome (%)	Atypical AR (%)	Typical AR (%)	Lost Acrosome (%)
Control	69.3 ± 2.40 ^b^	19.3 ± 0.88 ^a^	8.4 ± 1.45	3.0 ± 1.15
Extract 125 µg/mL	70.3 ± 1.20 ^b^	19.0 ± 0.58 ^a^	8.0 ± 0.58	2.7 ± 0.88
Extract 250 µg/mL	76.3 ± 1.45 ^a^	14.7 ± 1.45 ^b^	6.7 ± 0.33	2.3 ± 1.20
Extract 375 µg/mL	79.7 ± 0.88 ^a^	11.6 ± 1.20 ^b^	7.0 ± 0.58	1.7 ± 0.33

^a^ and ^b^: Values in the same column with different superscripts are significantly different at *p* < 0.05.

**Table 7 animals-11-02840-t007:** Effect of *T. fischeri* leaf extract supplementation on seminal antioxidant capacity and enzymatic activities (means ± SE, *n* = 3).

Concentration (mg/mL)	TAC (Mm/L)	H_2_O_2_ (nm/L)	LDH (U/mL)	AST (U/L)	ALT (U/L)
Control	0.57 ± 0.02	1.3 ± 0.09 ^a^	91.7 ± 8.67	63.3 ± 6.36	14.0 ± 2.00
Extract 125 µg/mL	0.62 ± 0.03	0.9 ± 0.04 ^b^	112.4 ± 13.25	60.0 ± 10.58	13.7 ± 0.88
Extract 250 µg/mL	0.67 ± 0.01	0.7 ± 0.09 ^b^	105.2 ± 21.64	60.0 ± 4.00	14.0 ± 2.00
Extract 375 µg/mL	0.69 ± 0.05	0.8 ± 0.05 ^b^	96.2 ± 15.91	60.0 ± 4.00	12.7 ± 0.67

^a^ and ^b^: Values in the same column with different superscripts are significantly different at *p* < 0.05. TAC = total antioxidant capacity, H_2_O_2_ = hydrogen peroxide, LDH = lactic dehydrogenase. AST = aspartate transaminase, ALT = alanine transaminase.

## Data Availability

The data that support the findings of this study are available from the corresponding author, W.A.K., upon reasonable request.

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
