# Peer review of "Antioxidant and Antiapoptotic Effects of a *Turraea fischeri* Leaf Extract on Cryopreserved Goat Sperm"

_animals, 2021, doi:10.3390/ani11102840_

Round 1
Reviewer 1 Report
According to the results in the manuscript, addition of TFLE into extender for semen cryopreservation had positive effect on viability and quality of goat sperms.
The usefulness of TFLE on semen cryopreservation seems to be clear, but I have several questions and comments on the manuscript before publication.
Table 1: Why compound no. 6 and 7 were picked up as Figure 1 and 2, even though any specific discussion about the compounds is included? I couldn’t find the reason.
Table 2: What is TPC assay? I couldn’t find the explanation.
Table 4: I understand that the rates of apoptotic sperms were lower in TFLE-treated semen possibly due to the antioxidant effect of TFLE. To the contrary, why the rate of necrotic sperms was lower in 375 ug/mL TFLE-treated semen? In addition, why the rate of necrotic sperms was higher in 125 and 250 ug/mL TFLE-treated semen than control?
Table 7 and line 291: The authors described that “Addition of TFLE also dose-dependently enhanced post-thaw TAC” but the result of statistical analysis is not indicated in TAC column of Table 7. Please add it.
Lines 290-293: The description “Table 7” is not included.
Are the benefits of TFLE on sperm cryopreservation expected only in goat semen or also in other species? Please discuss about the possibility.
Was the effect of TFLE on semen cryopreservation stronger or weaker compared with other antioxidants including plant extracts reported previously? Please discuss about this point.
Author Response
According to the results in the manuscript, addition of TFLE into extender for semen cryopreservation had positive effect on viability and quality of goat sperms. The usefulness of TFLE on semen cryopreservation seems to be clear, but I have several questions and comments on the manuscript before publication.
Thank you for your feedback and reviewing our manuscript. Your comments were valuable and have made edits to address your observations.
- Table 1: Why compound no. 6 and 7 were picked up as Figure 1 and 2, even though any specific discussion about the compounds is included? I couldn’t find the reason.
These compounds were tentatively identified based on their molecular weights and fragmentation pattern and we just showed them as examples from the extracts. However, we deleted the figures.
- Table 2: What is TPC assay? I couldn’t find the explanation.
TPC is total phenolic contents; however, we have explained all abbreviations in the table in the footnote of Table 2.
- Table 4: I understand that the rates of apoptotic sperms were lower in TFLE-treated semen possibly due to the antioxidant effect of TFLE. To the contrary, why the rate of necrotic sperms was lower in 375 ug/mL TFLE-treated semen? In addition, why the rate of necrotic sperms was higher in 125 and 250 ug/mL TFLE-treated semen than control?
We thank the reviewer for this insightful note. There are two main categories (live sperm with PI negative, which include AV-/PI- and AV+/PI − for live and early apoptotic sperms) and the dead sperm with PI positive (AV+/PI+, and AV-/PI+ for apoptotic and necrotic sperms). This was classified according to Peña et al. (2003) and also Anzar et al. (2002). So, we showed that 375 ug/mL TFLE supplementation increased live sperms in relative to the total necrotic and dead sperms.
F.J. Peña, A. Johannisson, M. Wallgren, and H. Rodrı́guez-Martı́nez, Assessment of fresh and frozen–thawed boar semen using an Annexin-V assay: a new method of evaluating sperm membrane integrity. Theriogenology 60 (2003) 677-689.
Anzar , L. He, M.M. Buhr, T.G. Kroetsch, K.P. Pauls, Sperm Apoptosis in Fresh and Cryopreserved Bull Semen Detected by Flow Cytometry and Its Relationship with Fertility, Biology of Reproduction, 66, (2002) 354–360, https://doi.org/10.1095/biolreprod66.2.354
4. Table 7 and line 291: The authors described that “Addition of TFLE also dose-dependently enhanced post-thaw TAC” but the result of statistical analysis is not indicated in TAC column of Table 7. Please add it.
We have clarified this issue in the text as follow: “Addition of TFLE also dose-dependently enhanced insignificantly post-thaw TAC and reduced significantly H2O2 (p < 0.05, Table 7)”.
- Lines 290-293: The description “Table 7” is not included.
We have mentioned Table 7 in the text.
- Are the benefits of TFLE on sperm cryopreservation expected only in goat semen or also in other species? Please discuss about the possibility.
We have discussed the possibility of using TFLE on sperm cryopreservation in other species.
Was the effect of TFLE on semen cryopreservation stronger or weaker compared with other antioxidants including plant extracts reported previously? Please discuss about this point.
We have discussed this point according to your comments.
Reviewer 2 Report
Dear Authors,
This is an interesting study and provides useful information on the effect of the addition of Turraea fischeri Leaf Extract on cryopreserved goat sperm. The manuscript presents a well-performed experimental design, including chemical composition of Turraea fischeri Leaf Extract and oxidant and antioxidant productions.
However, I have identified some minor issues that should improve the quality of manuscript and produce a revised manuscript suitable for publication:
The introduction is well written, but I suggest to specify if there are previous studies that reported the use of Turraea fischeri in reproduction and if this is the first study on the use of Turraea fischeri in the semen extender.
Line 129: Please modify in “number of sperm cells with unstained head area”
Line 134: Please specify the staining use for morphology evaluation.
Line 138: Please add modify PM in plasma membrane (PM)
Line 213: Please add at the end of the phrase “as shown in Table 3”
Table 7: In lines 291-292 the authors wrote “Addition of TFLE also dose-dependently enhanced post-thaw TAC and reduced H2O2 (p < 0.05).”, but you miss the superscripts for TAC’s values in the Table. Please insert the letters that shown the statistical differences.
The discussion is simple and quite straightforward, furthermore I suggest the authors to add the limitations of the study (e.g. number of ejaculates) and to purpose new perspectives.
I will be glad to review this manuscript again.
Regards,
Author Response
This is an interesting study and provides useful information on the effect of the addition of Turraea fischeri Leaf Extract on cryopreserved goat sperm. The manuscript presents a well-performed experimental design, including chemical composition of Turraea fischeri Leaf Extract and oxidant and antioxidant productions.However, I have identified some minor issues that should improve the quality of manuscript and produce a revised manuscript suitable for publication:
We thank the reviewer for the excellent review and the constructive comments.
- The introduction is well written, but I suggest to specify if there are previous studies that reported the use of Turraea fischeri in reproduction and if this is the first study on the use of Turraea fischeri in the semen extender.
To our knowledge, this is the first study to examine the effect of adding the Turraea fischeri leaf extract to semen extender. We have mentioned this in the aim of the study.
- Line 129: Please modify in “number of sperm cells with unstained head area”
We have modified it according to your comments.
- Line 134: Please specify the staining use for morphology evaluation
The dyes used for morphology evaluation are already in the section of viability.
- Line 138: Please add modify PM in plasma membrane (PM)
We have modified it according to your comments.
- Line 213: Please add at the end of the phrase “as shown in Table 3”
We have modified it according to your comments.
- Table 7: In lines 291-292 the authors wrote “Addition of TFLE also dose-dependently enhanced post-thaw TAC and reduced H2O2(p < 0.05).”, but you miss the superscripts for TAC’s values in the Table. Please insert the letters that shown the statistical differences.
We have clarified this issue in the text as follows: “Addition of TFLE also dose-dependently enhanced insignificantly post-thaw TAC and reduced significantly H2O2 (p < 0.05, Table 7)”.
- The discussion is simple and quite straightforward, furthermore I suggest the authors to add the limitations of the study (e.g., number of ejaculates) and to purpose new perspectives.
We have added a new perspective in the conclusion according to your comments.